# Adipocyte, Immune Cells, and miRNA Crosstalk: A Novel Regulator of Metabolic Dysfunction and Obesity

**DOI:** 10.3390/cells10051004

**Published:** 2021-04-24

**Authors:** Sonia Kiran, Vijay Kumar, Santosh Kumar, Robert L Price, Udai P. Singh

**Affiliations:** 1Department of Pharmaceutical Sciences, The University of Tennessee Health Science Center (UTHSC), 881 Madison Avenue, Memphis, TN 38103, USA; skiran@uthsc.edu (S.K.); vkumar7@uthsc.edu (V.K.); ksantosh@uthsc.edu (S.K.); 2Department of Cell and Developmental Biology, University of South Carolina, Columbia, SC 29208, USA; bob.price@uscmed.sc.edu

**Keywords:** adipocyte, miRs, inflammation, immune cells, metabolic dysfunction

## Abstract

Obesity is characterized as a complex and multifactorial excess accretion of adipose tissue (AT) accompanied with alterations in the immune response that affects virtually all age and socioeconomic groups around the globe. The abnormal accumulation of AT leads to several metabolic diseases, including nonalcoholic fatty liver disorder (NAFLD), low-grade inflammation, type 2 diabetes mellitus (T2DM), cardiovascular disorders (CVDs), and cancer. AT is an endocrine organ composed of adipocytes and immune cells, including B-Cells, T-cells and macrophages. These immune cells secrete various cytokines and chemokines and crosstalk with adipokines to maintain metabolic homeostasis and low-grade chronic inflammation. A novel form of adipokines, microRNA (miRs), is expressed in many developing peripheral tissues, including ATs, T-cells, and macrophages, and modulates the immune response. miRs are essential for insulin resistance, maintaining the tumor microenvironment, and obesity-associated inflammation (OAI). The abnormal regulation of AT, T-cells, and macrophage miRs may change the function of different organs including the pancreas, heart, liver, and skeletal muscle. Since obesity and inflammation are closely associated, the dysregulated expression of miRs in inflammatory adipocytes, T-cells, and macrophages suggest the importance of miRs in OAI. Therefore, in this review article, we have elaborated the role of miRs as epigenetic regulators affecting adipocyte differentiation, immune response, AT browning, adipogenesis, lipid metabolism, insulin resistance (IR), glucose homeostasis, obesity, and metabolic disorders. Further, we will discuss a set of altered miRs as novel biomarkers for metabolic disease progression and therapeutic targets for obesity.

## 1. Introduction

Chronic inflammation associated with obesity affects people around the world, especially in western regions including the USA and Canada. According to the World Health Organization, approximately 38.2 million children less than 5 years old were obese in 2019. Global statistics show obesity is a risk component contributing to the development of severe metabolic disorders such as nonalcoholic fatty liver disorder (NAFLD) [1], cancer [2], type 2 diabetes mellitus (T2M) [3], and cardiovascular disorders (CVDs) [4]. Obesity alters preexisting adipocyte hypertrophy, preadipocyte differentiation, macrophages, T-cell infiltration at the adipose site, the release of inflammatory cytokines, and increases insulin resistance (IR) [5]. In ATs, cytokines and chemokines are released by both adipocytes and immune cells [6], and the dysfunction of these AT-derived-cytokines and chemokines may result in hypoxia, and excess secretion of free fatty acids (FFAs) [7]. Moreover, the infiltration of immune cells and the excessive accumulation of macrophages may also contribute to FFA release [8]. In obese individuals, adipocytes increase in quantity and mass and change their immunological profile. These cell deaths trigger the release of metabolites, which polarize the immune response toward a pro-inflammatory T-helper 1 (Th1) state. The excess FA in AT acts as an energy source for the proliferation of cancer cells and the development of tumors [9]. Furthermore, the up-regulation of various inflammatory genes in immune cells induces a pro-inflammatory immune response in ATs, glucose tolerance, and IR, the main factors correlated with metabolic syndrome [10].

Innate lymphoid cells appeared as key element in obesity and immune response. Interestingly, ILC2 have been reported in gut mucosa and lymphoid clusters associated with fat [11]. IL-5, that is obtained by ILC2, is essential for the stimulation of eosinophil and their migration to AT, while IL-13 promote macrophage activation [12]. Together, these ILC2-derived cytokines help in macrophage homeostasis and play a protective role in obesity-induced metabolic disorders. Meanwhile, the deficiency of ILC2 elevates adipocity and IR in HFD- mice [13]. In AT, the ILC2 are elevated by IL-33 and IL-23 signals [14]. ILC2 limit the inflammation and help to maintain metabolic homeostasis adipose-resident ILC1 promote disease progression such as IR [15]. Hence, it is shown that ILC2 has a protective role against obesity while ILC1 has adverse effects. Metabolic syndrome may also enhance the inflammatory mediator secretion such as IL-6 [16], leptin [17], TNF-α [18], resistin [19], and monocyte chemoattractant protein-1 (MCP-1) [20]. It has been shown that in AT, the number of T-cells is elevated in both mice and humans [21]. Additionally, both CD4^+^ and CD8^+^ T-cells are actively involved in obesity-associated inflammation (OAI) in white adipose tissue (WAT). The pro-inflammatory Th1 cells in WAT of mice and humans in obese conditions recruit macrophages to induce an inflammatory microenvironment in the vascular adipose tissue (VAT). Interestingly, both T-cells and macrophages work together and help the body fight against foreign stimuli, repair tissues, and remove dead cells. Both macrophages and T-cells work synergistically to build an effective immune response through different immunological mechanisms, including phagocytosis and cytokine or chemokine secretion [22]. Sometimes, the excessive accumulation of inflammatory cytokines occurs, which might cause chronic inflammation and cause various diseases such as obesity and aging [23]. In the last decade, various studies have highlighted macrophage accumulation in ATs as the main source of inflammation aiding in meta-inflammation, and miRNAs are a key component of these signaling pathways [24,25]. Hence, the current review is designed to discuss the cross-talk between AT-associated adipocytes, immune cells, and miRs in regulating OAI and metabolic dysfunction.

## 2. MicroRNA—An Exciting Discovery

MicroRNAs (miRs) are small non-coding RNA molecules produced by almost all types of eukaryotic cells and consist of approximately 20–22 nucleotides (NTs) [26]. miRs regulate multiple genes by binding to target mRNAs, thereby controlling the stability and translation of protein-coding mRNAs [27,28]. The number and expression of miRs varies with species and depends on their complexity. Humans have 2000 to 3000 miRs, while mice have over 1500 miRs, with some being tissue and organ-specific [29]. The synthesis of miRs begins in the cell nucleus, where primary miRs are transcribed by RNA polymerase II. These primary miRs are transferred to the cytoplasm by Ran GTPase and exportin-5 (XPO5) (Figure 1). The primary miRs produce miRNA with the aid of type III endoribonuclease DICER in connotation with RNA-binding proteins. Another pathway for the biogenesis of miRs has been reported as DICER-independent [30] and miRs serve as a signature of cell identification. The miRs are expressed in different organs of the body by different cell types; however, it has been shown that miR-122 is greatly expressed in the liver [31]. Similarly, miR-133a, [32] miR-133b, [33] miR-208a, [34] miR-208b, miR-499, [35] miR-486, [36] and miR-1 [37] help in the proliferation and differentiation of skeletal muscles, are expressed in muscle cells and are known as myomiRs [38] About 50% of brain tissue miRs consist of miR-124 and miR-9 [39,40].

miRs play a key role in cellular interaction via a mechanism involving protein carriers and vesicle trafficking. To this end, Killian et al. identified several miRs inside the extracellular vesicles, their secretion, and uptake by donor and receiving cells, respectively, followed by the release of their cargo in receiving cells [41,42]. Later on, Cortez and his coworker confirmed the presence of miRs in body fluids and also correlated their level to disease development [29]. The extracellular transport of miRs occurs by two routes: (a) active transport with the aid of extracellular vesicles [43] and (b) as a part of protein–miRNA complexes [44]. It has been shown that sometimes miRs also leak from damaged cells [45] to mediate a physiological response including growth regulation, immune responses, and reactions toward external stimuli.

## 3. Role of miRs in AT Inflammation

miR dysregulation in AT causes inflammation that is directly linked with obesity. Further, these dysregulated miRs are also closely associated with various obesity-related metabolic diseases. Recently, 21 miRs expressed differently in lean and obese people have been identified in epididymal AT [46]. Another group has analyzed human subcutaneous AT by performing miRNA arrays and found that 50 of 799 miRs show a significant difference in both lean and obese individuals [47]. It has been shown that out of these 50 miRs, around 17 well correlate with BMI [47]. The expression of miR-17–5p and miR-132 causes a decrease in omental fat, hence its circulation is also minimized in obese individuals [48]. Another recent study highlighted approximately 11 adipocyte-linked miRs whose concentration is downregulated in obese individuals [49]. Due to the very close association between obesity and inflammation, it is highly suggested that the dysregulation of miRs in inflammatory adipocytes is also closely associated with obesity-induced inflammation. To support this hypothesis, a few miRs have been reported that play a critical role in the inflammatory state of AT. miR-221 and miR-222 expression correlates with both TNF-*α* and aminopeptidase N-1; however, the effect of miRNA on these adipokines is still a question [50]. Moreover, miR-132, which is downregulated in obese individuals, activates NF-κB and the transcription of MCP-1 and IL-8 [51]. Brichard et al. have identified that adiponectin is regulated by miR in WAT and among these miRs, it is observed that miR883b-5p is upregulated in adiponectin and downregulated in obese individuals [52]. Together, these studies suggest that AT miRs regulate adipokines either directly by focusing on the inflammatory adipokines or indirectly by the initial regulation of intermediate components that control adipokines in the next stages of inflammation.

T-cells develop in the thymus and then move to their corresponding organs to attain tissue or organ-specific characteristics. Recent studies have shown that miRs are actively involved in the maturation and differentiation of T-cells [53,54]. Among these miRs, most are derived from adipocytes and are actively involved in T-cell recruitment and inflammation. miR-326 participates directly in the polarization of Th1 cells towards Th17 cells which endorse the inflammation of AT through releasing IL-17; hence, miRs serve as a marker for Th17 in inflamed AT [55]. These studies highlight that miRs expression in AT is directly associated with T-cell inflammation, the process stimulated by AT during the onset of obesity. Beside macrophages and T-cells, it is reported that B-cells miRs also play a key role in obesity. For example, miR-150 is studied to modulate AT function by B-cell activation and correlation with other immune cells [56].

## 4. Role of miRs and Oxidative Stress in Obesity and Its Associated Diseases

The imbalance between the production of free radicals and anti-oxidant is termed as oxidative stress (OS). Due to access supply of glucose and free fatty acid (FFA) toward mitochondria, an increase in ROS and OS is observed in obese individuals, and it has been shown that obesity is related to an enhanced concentration of free radicals in blood [57] (Figure 2). Further, various enzymes such as lipoxygenases, cytochrome P450 (CYP), nitric oxide (NO) synthases and oxidases is generated by ROS and work as inflammatory mediators [58]. ROS generation by NOXs, (an enzyme that helps to generate ROS from oxygen) is associated with OS and helps in the regulation of some of the miRs. Intriguingly, it is observed that the level of miR-26a is downregulated in mice fed on HFD as compared to ND [45]. However, it has been shown that miR-26a causes the downregulation of protein kinase C δ (PKCδ), which is involved in the production of ROS [59]. Therefore, miR-26a could play a role against hepatosteatosis. The overexpression of miR-34a enhances the OS in NAFLD [60]. The lower level of NAD+ due to miR-34a targeting nicotinamide phosphoribosyltransferase (NAMPT) affects the activity of silent information regulator 1 (SIRT1), suppressing the lipid oxidation and promoting the inflammation of lipid [61]. The upregulation of miRs-217 is observed in atherosclerotic lesions, and this miRs is known to target SIRT1, a regulator of metabolic disorders [62]. The deacetylase SIRT1 and forkhead box-containing protein, O subfamily1 (FoxO1), enhance the activity of endothelial nitric oxide synthase [61,63]. The correlation between SIRT1 and miRs might serve as a new highlight in atherosclerosis regarding endothelial nitric oxide synthase activity. Sun et al. reported that miR-155 and simvastatin serve as anti-atherosclerosis, as simvastatin downregulate the expression of miR-155 through affecting the mevalonate-geranylgeranyl-pyrophosphate-RhoA signaling pathway [64]. In another study, Yang et al. highlighted that the expression of the miR-155-target protein, mammalian sterile 20-like kinase 2 (MST2) is enhanced in the arteries of miR-155 knockout mice. The miR-155 suppressed MST2 and stimulated the pathway of the extracellular signal-regulated kinase (ERK); as a result, the OS response was stimulated [65]. Together, these findings highlighted the fact that obesity and OS are closely related; hence, serve as a key mediator in the progression of obesity-associated metabolic disorders [66]. Although, various antioxidant compounds are reported to defend against obesity associated-disease; however, considering the miRs’ ability to target specific proteins, it is speculated that these miRs could serve as therapeutic targets to alleviate OS in obesity-related metabolic disorders.

## 5. Role of miRs in Metabolic Organ Cross-Talk

In recent years, several reports have supported the fact that circulating and tissue-derived miRs facilitate intracellular communication. The role of miRs in physiological processes of metabolic organ crosstalk has also been acknowledged as an endocrine and paracrine messenger. The role of miRs as endocrine messengers takes place in three different ways: first, hormone production and the response of target cells are regulated by miRs. As an example, miR-378a-5p targets a member of the transforming growth factor-β family (nodal) [55]. Second, cytokines regulate miRs. For example, IL-6 stimulates STAT3 to activate the transcription of miR-21 [67] and miR-181-b1 genes [68,69]. Third, miRs show hormone-like properties and facilitate cell to cell interaction. Host cells secrete miRs and they are taken up by receipt cells to perform their activities [70]. Hence, these studies show that miRs play a key role in the crosstalk between different organs to maintain their homeostasis (Figure 3). Below, we discuss some miR crosstalk with metabolic organs in more detail.

## 6. AT Derived miRs

Obesity is identified as an excessive accumulation of WAT and brown AT (BAT) [71]. WAT is characterized as an energy storage site and BAT generates heat by utilizing the stored energy via thermogenesis phenomena. BAT is known as the secretary type of AT and releases adipokines and miRs [72]. Adipokines help in the maintenance of energy homeostasis in the whole body. AT-derived circulating miRs are newly identified adipokines [73]. To this end, patients suffering from lipodystrophy have low circulating miRs as compared to healthy people [74]. DICER is studied as main miRNA biogenesis enzyme. Recently, Brandao et al. reported that exercise training stimulates the DICER in AT of both mice and human. Moreover, miR-203-3p is upregulated as a result of exercise training. Together, DICER-miR-203-3p is upregulated as a result of exercise, which ultimately helps the body in metabolic function [75]. Mudhasani et al. studied the genetic role of miRNA in the regulation of adipogenesis and DICER requirement in WAT and BAT formation [76]. Another study was conducted on DICER-deficient mice, indicating a defective miRs processing and lipodystrophic phenotype. Hence, these findings support the fact that AT microenvironments contribute to circulating miRs expression and function. Most of these miRs are released by adipocytes and their expression level is directly related to the severity of obesity. A DICER-deficient mouse was transplanted with wild-type mouse AT and it was observed that circulating miRs expression was reconstituted to normal and an improved glucose tolerance level was obtained. Most of these AT-derived circulating miRs were also associated with body mass index (BMI), waist-to-height ratio, percent fat mass, and plasma adipokine levels in both humans and mice. Particularly, it has been shown that the overexpression of miR-142-3p and miR-140-5p is reduced following bariatric surgery [77]. The major concentration of AT consists of adipocytes; however, fibroblasts, lymphocytes, adipocyte progenitor cells, and macrophages are also resident in the AT. These resident cells help in AT regulation and play a key role in the development of inflammation, IR, and secretion of adipokines. However, miRs derived from adipocytes act as the main regulator to energy homeostasis, and AT acts as the main source of circulating miRs.

## 7. miRs as A Bridge between Adipocytes and AT Macrophages

Macrophages are an essential component of the innate immune system and are important for mediating the host defense against inflammation. Macrophages may have classical M1 (pro-inflammatory) or M2 (anti-inflammatory) phenotypes. The crosstalk between adipocytes and AT macrophages (ATMs) is quite challenging due to metabolic complications in obese individuals. AT in lean individuals is populated by M2 macrophages that express F4/80, CD11b, CD206, and CD301 in mice [78]. M2 macrophages maintain AT homeostasis and insulin sensitivity [79]. AT of lean mice is populated mostly by IL-4 [80]. Classical M1 macrophages are highly expressed in obese mice and secrete pro-inflammatory factors, such as IL-6, TNF-α, and nitric oxide (NO) [81,82]. For example, the percentage of M1 macrophages increases to 50% in AT of obese people as compared to lean subjects, who have only 5% of these immune cells [83]. ATMs mainly facilitate efferocytosis, regulate adipocyte lipolysis, and produce anti-inflammatory cytokines in lean conditions. However, in obese individuals the anti-inflammatory phenotype switches towards a pro-inflammatory phenotype and the secretion of excess cytokines such as IL-6 and TNF-α, resulting in AT low-grade chronic inflammation, is present. This low-grade inflammation also results in IR and the development of diabetes. In an obese microenvironment, both adipocyte and ATMs secrete microvesicles and exosomes containing mi-RNAs in blood vessels that affect glucose homeostasis. AT-derived adipocytes release more miRs containing exosomes in obese mice as compared to lean mice [84]. Lee et al. conducted bioinformatics analysis of RNA and identified that miR-10a-5p is a key regulator of inflammation in ATMs and reported that a high-fat diet reduces miR-10a-5p levels in ATMs, and the treatment of mice with a miR-10a-5p mimic inhibits pro-inflammatory responses and enhances glucose tolerance [85].

miR-29a levels increase in obese ATM-derived exosomes, which might result in IR [86]. However, the knockout of miR-29a in obese ATM exosomes minimizes this factor. Hence, miR-29a could serve as an obesity-associated type 2 diabetes marker [87]. It has been also shown that miR-30e-5p is upregulated in ATMs from high fat diet (HFD) mice when treated with AM251, an antagonist of cannabinoid receptor 1, while its target, DLL4 is downregulated [88]. Hence, miR-30e can serve as a biomarker in cardiometabolic disorders. Later, it was reported that HFD-induced obesity downregulates miR-30 and acts as a regulator of pro-inflammatory ATM also be developed as a marker for obesity-induced inflammation [25]. miR-99a was reported as a negative expression of obesity, M2 macrophages show the overexpression of miR-99a, while M1 macrophages show a decreased expression, indicating the relationship between M2 phenotype and miR-99a [89]. To this end, TNFα was recognized as a direct target of miR-99a and can decrease the inflammation of AT to improve insulin sensitivity. Another miR-155 is overexpressed in obese ATMs and targets peroxisome proliferator-activated receptors (PPARγ). These miRs enter in insulin-secreting pancreatic islets of β cells through endocrine or paracrine regulation via robust effects on cellular insulin action to maintain glucose homeostasis [90]. miR-34a is identified as a vital mediator and attack on adipose-resident M2 macrophage to stimulate obesity and chronic inflammation [91]. Taken together, these circulating and tissue-derived miRs facilitate the cross-talk between adipocytes and ATMs to sustain obesity and inflammation.

## 8. AT and Skeletal Muscle miRs

miRs also mediate the communication between skeletal muscles and ATs. For example, miR27a facilitates cross-talk between AT and skeletal muscle [92]. Both obese and T2DM patients have higher circulating levels of miR27a than healthy/lean individuals. Additionally, miR27a levels increase in obese patients through a different mechanism; one of them may include a decrease in leptin levels. We can speculate that leptin may affect obesity via regulating miR27a circulating levels, but this needs further investigation. Additionally, miR27a secreted by adipocytes in ATs may enhance OAI through regulating the macrophage polarization to their pro-inflammatory phenotype M1-macrophages [93]. The increased circulating levels of miR27a may increase the predisposition of obese people to IR through inhibiting the PPAR-γ signaling pathway. For example, the antidiabetic role of PPAR-γ in ATs is well established, along with other major glucose level regulating organs, such as the liver and pancreas [94,95]. The exosome trafficking of miR-27a helps skeletal muscle cells uptake these miRNAs. The palmitate-treated 3T3-L1 adipocytes derived from exosomal miR-27a inhibit PPAR-γ and result in IR in C2C12 cells [96]. Additionally, M1 macrophages in AT microenvironment also release miR-155 that decreases the PPAR-γ coactivator 1α (PGC1α), resulting in IR and predisposition to T2DM [90]. PGC1α has a proinflammatory role as it enhances lipid peroxidation (LPO) within skeletal muscle [97]. Hence, miRs may have both pro and anti-inflammatory action in establishing crosstalk between AT and skeletal muscles.

## 9. miRs in AT–Pancreas Crosstalk

To date, nothing has been reported that supports crosstalk between AT and the pancreas. However, some circulating miRs derived from AT are shown to regulate pancreatic β-cells [98]. To this end, miR-132 is reported as a promoter of β-cell proliferation, and its level is reduced in obese individuals and AT [99]. The higher expression of miR-132 stimulates the proliferation of pancreatic α-cells that leads to the development of pancreatic carcinoma. Moreover, the overexpression of BAT-derived miR-92 in INS-1 cells (a rat insulinoma cell line) reduces insulin by the downregulation of polypyrimidine tract binding protein 1 (PTBP1) [99]. The serum level of miR-146b and miR-15b is enhanced in obese and T2DM individuals, while the overexpression of miR-146b and miR-15b could lower the secretion of insulin in mouse pancreatic MIN6 cells line [100]. These findings highlight the importance of miRs as an extra shield of negative gene directive and their contribution to pancreatic β-cell maintenance and IR. However, further studies need to be conducted for any prudent conclusion to facilitate AT–pancreas crosstalk.

## 10. AT-Cardiovascular System miRs

Recent evidence supports the fact that circulating and tissue-derived miRs facilitate the progression of cardiovascular diseases (CVDs) [101]. An atherosclerosis mouse model (Apoe^-/-^) was administrated with exosomes derived from AT of mice that were fed a HFD, and these mice showed an exacerbation of atherosclerosis that might be due to exosomal contents, including miRs [102]. Further, miR-29a and miR-194 show direct correlation between obesity and cardiac dysfunction and reduced mitochondrial function of primary cardiomyocytes in mice [103]. The cardiac dysfunction is attenuated in HFD-fed mice by using miR-29a mimics or miR-194 inhibitors [104]. Further, miR-410-5p induces cardiac fibrosis in mice and is overexpressed in HFD-fed mice. Taken together, miRs can be developed as an attractive marker to investigate the communication between AT and cardiovascular tissues [105].

## 11. Endocrine Function in the Liver by miRs

It has been shown that circulating miRs released from AT facilitates crosstalk with the liver and function as an endocrine factor to mediate liver function. miR-122 is highly expressed by hepatocytes and is 75% of the total miRs present in liver tissue [106]. Further, miR-122 is essential for lipid metabolism and shows antitumorigenic and anti-inflammatory properties [107]. The overexpression of miR-122 has been noticed in patients suffering from nonalcoholic steatohepatitis [108]. Therefore, it is hypothesized that miR-122 released from the liver of healthy individuals is decreased, but in obese individuals their percentage is increased as AT also secretes an excess amount of miR-122 to maintain the function of the liver and decrease the chances of liver disease. A BAT-derived miR-99b also helps in the regulation of hepatic metabolism and is identified as a suppresser of fibroblast growth factor 21 (FGF21) in mouse liver [109]. These BAT-derived exosomal miRs block the synthesis of fibroblast growth factor 2 (FGF2), while miR-155 aids in glucose homeostasis in the miR-155 knockout mouse (KO) model [110]. These studies highlight the role of AT-derived miRs in the liver.

Taken together, the above analysis highlights the fact that miRs secreted from AT act differently in response to different external stimuli. Once these miRs enter the blood, they communicate with different organs such as the muscles, heart, liver, and pancreas to maintain metabolic homeostasis. Table 1 is a list of some miRs reported from various diseases.

## 12. Extracellular miRs as Disease Biomarkers

Recent studies suggest that extracellular miRs serve as ideal disease biomarkers in head and neck squamous cell carcinoma [131], inflammation [132], and pancreatitis [133]. These miRs can be detected in body fluids, such as blood [134], by applying a sensitive, cheap, and simple assay. Indeed, fluctuation in the expression of these circulating and tissue miRs concomitant with different diseases, including obesity [135], T2DM [136], cardiac disorder [65], cancer [137], aging [138], and neurodegenerative disorders such as Huntington’s disease (HD), Alzheimer’s disease (AD), and Parkinson’s disease (PD) has been identified [139].

## 13. miRs-Based Therapeutics

Various miRs have been identified as attractive therapeutic markers for neurocognitive disorders [140], cancer [141], metabolic syndrome [142], and autoimmune diseases [143,144]. miRs-based therapies are classified as miRs mimics and anti-miRs oligonucleotides. In 2018, the FDA approved patisiran as the first-ever RNA-based drug used for the treatment of a neurodegenerative disorder (such as Alzheimer’s and Parkinson’s diseases) [145]. Miravirsen is an anti-miR-122 drug that has entered phase II trials to cure hepatitis virus (HCV) infection [146]. It has been shown that Miravirsen inhibits the biogenesis of miR-122 and suppresses HCV infection. Moreover, this drug has a negligible toxic effect and viral resistance. Other miR-34 and miR-16 mimics have been used to cure gastric cancer [147] and solid tumors, while cutaneous T-cell lymphoma is being treated by anti-miR-155 [148]. Anti-miR-92a has entered phase I clinical trials for wound healing [149] research, while miR-29 mimic has entered phase II clinical trials for keloid patients. Surprisingly, various obesity-linked miRs have been reported to have clinical potential, but only anti-miR-103 and anti-miR-107 have reached phase I trials to treat T2DM patients.

miR derived from AT helps to maintain the energy homeostasis of various physiological processes within the body. Interestingly, it has been observed that deletion of the DICER encoding gene of AT in mice results in the depletion of miRs and causes metabolic disorders [150], suggesting that these miRs may serve as therapeutic targets for metabolic syndrome. Keeping these properties in mind, this tissue-derived miRs has served as a therapeutic for many diseases and has been investigated in several disease animal models for their mechanism of action. It has been shown that the removal of miR-155 in mice can minimize the HFD-induced IR and glucose intolerance [151]. Carlos et al. observed that when a lean mouse-derived exosome is transfected with artificial mimics of miR-192, miR-122, miR-27a-3p, and miR-27b-3 that are overexpressed in obese mice and injected again in lean mice fed with a normal diet, glucose intolerance is observed [152]. The inhibition of miR-143-3p via the injection of anti-miR-143-3p in mice regulates the insulin-like growth factor 2 receptor (IGF2R) and prevents obesity-induced IR [153], and Aimin et al. reported that miR-34a–KO mice show HFD obesity-induced IR, systemic inflammation, and glucose intolerance [91]. Hence, miR-27a-3p, miR-122, miR-192, miR-143-3p, miR-155, and miR-34a can also act as therapeutic targets for metabolic syndrome and obesity. BAT adipogenesis is inhibited by miR-27 and miR-34 in mice, both miRs levels are elevated in obese individuals. Therefore, circulating and tissue-derived miRs are considered as a therapeutic target in obesity and obesity-related diseases and serve as a biomarker for pancreatic islet function, IR, and chronic inflammation. To date, the miRs entered in clinical trials for the treatment of various diseases are listed in Table 2.

## 14. Future Prospective and Conclusions

As discussed in this review, obesity is the hallmark of metabolic syndrome, characterized as an abnormal accumulation of AT and immune cells. Thus, increasing the knowledge of AT biology is fundamental to further understand the core functions of ATMs in obesity. miRs are emerging regulators of host homeostasis and innate immune response, including the immune homeostasis. Hence, understanding miRs synthesis, regulation, secretion, and impact on adjacent or distant cells (as miRs laden in extracellular vesicles) is crucial for different human conditions associated with altered metabolism and the immune status, including the obesity that has taken the shape of the epidemic in the modern world. In recent years, various designs and strategies have been proposed for the delivery of miRs mimics or antagomiRs. These have great potential to open avenues for therapeutics targeting obesity-associated altered metabolism and inflammatory status responsible for inducing insulin resistance (IR), T2DM, and atherosclerosis. We warrant further well-planned studies on the crosstalk between adipocytes, its derived factors, T-cells, and macrophages to give a clear direction to the obesity field.

## Figures and Tables

**Figure 1 cells-10-01004-f001:**
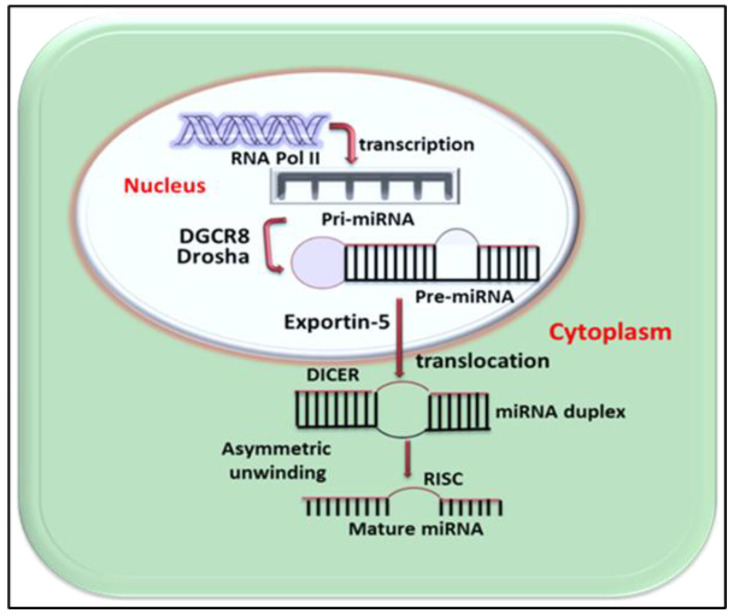
The schematic illustration of miRNA biogenesis and mechanism of action: in the nucleus of cells pri-miRNA are transcripted from DNA with help of RNA polymerase. The pri-miRNA is then cleaved by Drosha enzyme to pre-miRNA and moved into the cytoplasm by exportin-5 activity. The pre-miRNA is matured to miRNA via DICER. The mature strand of miRNA is integrated into the RNA-induced silencing complex (RISC) which inhibits the translation of the complementary mRNA.

**Figure 2 cells-10-01004-f002:**
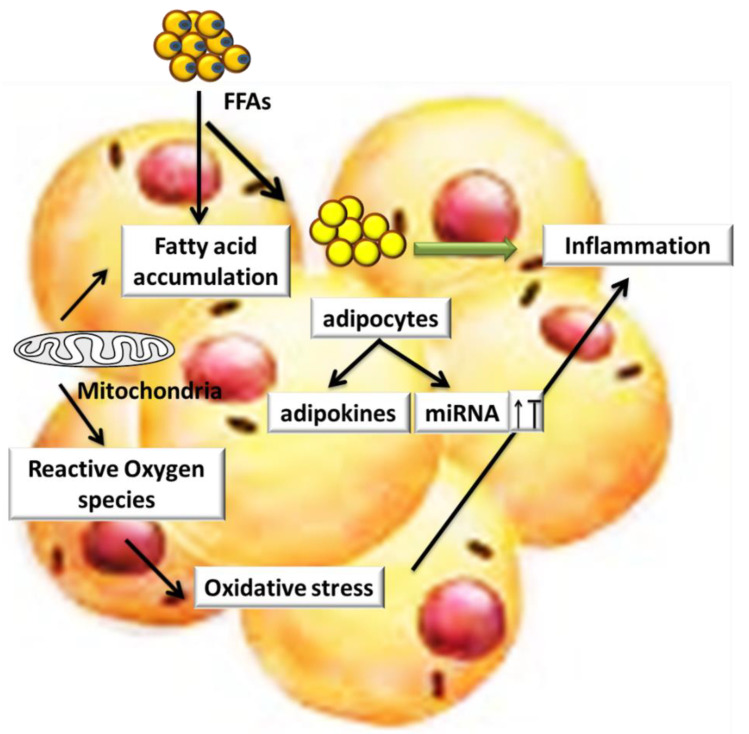
ROS and Obesity: In obese individuals, access supply of free fatty acid (FFA) toward mitochondria results in an increase in the production of ROS and OS, which leads toward inflammation; adipokines and miRs are also associated with OS and inflammation.

**Figure 3 cells-10-01004-f003:**
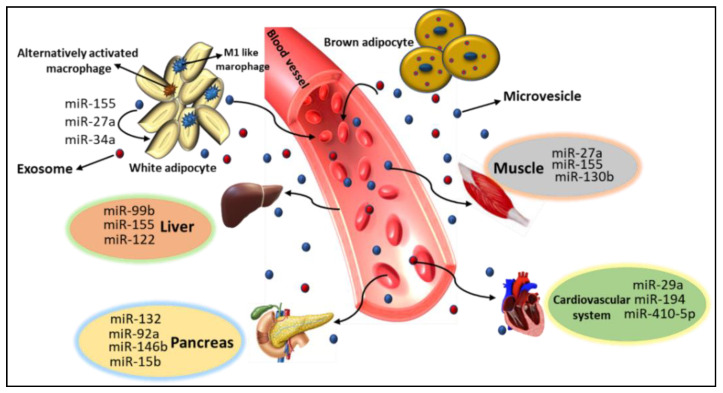
Crosstalk between AT-derived miRs and metabolic organs: microvesicles and exosomes loaded with miRs are secreted from AT into blood stream and distributed to different organs including liver, pancreas, muscles, and cardiovascular system to play their functions.

**Table 1 cells-10-01004-t001:** miRs expression in various inflammatory diseases.

Sr. No	Disease	miRs	References
1.	Alzheimer’s disease (AD)	let-7d	[111]
		miR-342	[112]
		miR-125b	[113]
2.	T2DM	miR-146a	[114,115]
		miR-223	[116]
		miR-142-3p	[117]
		miR-126	[118]
		miR-375	[119]
		miR-155	
3.	Obesity	miR-138	
		miR-122	[120]
		miR-142-3p	[121]
		miR-221	[122]
		miR-145	[123]
4.	Cardiac disorders	miR-92a	[124]
		miR-223	[125]
		miR-126	[126]
		miR-320a	[127]
5.	Parkinson’s disease (PD)	miR-29a/c	[128]
		miR-19b	[129]
		miR-214	[130]
		miR-133b	[60]

**Table 2 cells-10-01004-t002:** List of miRs in clinical trial for treatment of various diseases.

Sr. No.	miRNAs	Diseases	Drug	Ref.
1	anti-miR-155	Cutaneous T and B-cell lymphoma	MRG-106	[154]
2	anti-miR-122	Hepatitis C virus infection	Mirvirasen, RG-101	[155]
3	anti-miR-103/107	T2DM with nonalcoholic fatty liver disease	RG-125/AZD4076	
4	miR-34 mimic	solid tumors	MRX34	[156]
5	miR-16 mimic	Malignant pleural mesothelioma, non-small-cell lung cancer	MesomiR-1	[157]
6	miR-29 mimic	Scleroderma	MRG-201	[158]

## Data Availability

Not Applicable.

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
