# Peer review of "Adipocyte, Immune Cells, and miRNA Crosstalk: A Novel Regulator of Metabolic Dysfunction and Obesity"

_cells, 2021, doi:10.3390/cells10051004_

Round 1

Reviewer 1 Report

The manuscript by Sonia Kiran et al., addresses an interesting and current problem about the role of miRNA as a novel regulator of metabolic dysfunction. The review is well written and very interesting. I kindly suggest to highlight a little the “link” between obesity and of oxidative stress. In fact, it is well known that chronic inflammation is associated with the obesity, but at the same time it is well known that chronic inflammation is associated with oxidative stress too. I only suggest discussing more this aspect. A very interesting manuscript written by Metere et al., “The effect of sleeve gastrectomy on oxidative stress in obesity” treats the role of oxidative stress in obesity showing as the weight loss (induced by bariatric surgery) ameliorate the oxidative stress pattern. I suggest to integrate your review with some useful concepts you could find in that manuscript, that I think should be cited in a review about the obesity.
I congratulate the authors on this exciting review and with the exception of this supplement that I request, I observe no other major criticisms.

Author Response

Reply to Reviewers 1 Comments:

I only suggest discussing more this aspect. A very interesting manuscript written by Metere et al., “The effect of sleeve gastrectomy on oxidative stress in obesity” treats the role of oxidative stress in obesity showing as the weight loss (induced by bariatric surgery) ameliorate the oxidative stress pattern. I suggest to integrate your review with some useful concepts you could find in that manuscript, that I think should be cited in a review

 about the obesity.

Response: We thank the reviewer for his valuable and insightful suggestion to strengthen the review article. In the revised review, we have added a paragraph on the link between obesity and oxidative stress. Page 4 line 155-188

Role of miRNA and oxidative stress in obesity and its associated diseases

The imbalance between the production of free radicals and anti-oxidant is termed as oxi-dative stress (OS). Due to access supply of glucose and free fatty acid (FFA) toward mitochondria an increase in ROS and OS is observed in obese individuals, and it has been shown that obesity is related to an enhanced concentration of free radicals in blood [57] (Fig. 2). Further, various enzymes such as lipoxygenases, cytochrome P450 (CYP), nitric oxide (NO) synthases and oxidases is generated by ROS and work as inflammatory meditators [58]. ROS generation by NOXs, (an enzyme that helps to generate ROS from oxygen) is associated with OS and helps in the regulation of some of the miRs. Intriguingly, it is observed that level of, miR-26a is downregulated in mice fed on HFD as compared to ND [45]. However, it has been shown that miR-26a cause the downregulation of protein kinase C δ (PKCδ), which is involved in the production of ROS [59]. Therefore, miR-26a could play a role against hepatosteatosis. The overexpression of miR-34a enhances the OS in NAFLD [60].  The lower level of NAD+ due to miR-34a targeting nicotinamide phosphori-bosyltransferase (NAMPT), affects the activity of silent information regulator 1 (SIRT1), suppressed the lipid oxidation and promoting the inflammation of lipid [61]. The upregulation of miRs-217 is observed in atherosclerotic lesions, and this miRs is known to tar-get SIRT1, a regulator of metabolic disorders [62]. The deacetylase SIRT1 and forkhead box-containing protein, O subfamily1 (FoxO1), enhance the activity of endothelial nitric oxide synthase [61, 63]. The correlation between SIRT1 and miRs might be serve as a new highlight in atherosclerosis regarding endothelial nitric oxide synthase activity. Sun et. al reported that miR-155 and simvastatin serve as anti-atherosclerosis, as simvastatin downregulate the expression of miR-155 through affecting the mevalo-nate-geranylgeranyl-pyrophosphate-RhoA signaling pathway [64].  In another study Yang et al highlighted that the expression of the miR-155-target protein, mammalian sterile 20-like kinase 2 (MST2) is enhanced in the arteries of miR-155 knockout mice. The miR-155 suppressed MST2 and stimulated the pathway of the extracellular sig-nal-regulated kinase (ERK), as a result, the OS response is stimulated [65]. Together these findings highlighted the fact that obesity and OS are closely related hence, serve as a key mediator in the progression of obesity-associated metabolic disorders [66]. Although, various antioxidant compounds are reported to defend against obesity associated-disease, however considering the miRs ability to target specific proteins, it is speculated that these miRs could serve as therapeutic targets to alleviate OS in obesity-related metabolic disorders.

Reviewer 2 Report

In their review, Kiran et al., provide a summary of the literature on the cross talk between miRNA and organs/immune cells in the context of obesity. The authors conclude in proposing some intervention strategies with miRs-based therapies. The review is properly written but some aspects need to be better explain. I have recommendations that may help to further improve clarity.

- Page 2, lines 51 to 67 : It would be interesting for the readers to add some aspects of macrophages regulation during obesity by the recently discovered family of innate lymphoid cells.

- Page 2/3, lines 80 to 84 : It could informative to mention the targets of the miRs cited by the authors and their precise role(s)

- Page 4, line 125 to 133 : I would just mention here that the role of miR and macrophages will be detailed later in the review. Are there evidences on the role of miR in other immune cell subsets during obesity-associated inflammation ?

- Page 4 : line 141 : Please replace “hormones” by “cytokines”

- Page 4 line 142 : Please add example(s) with references.

It would be interesting to summarize (and/or to describe deeper) what it is known about DICER deficient mice and metabolic diseases (Mudhasani et al., J Cell Physiol 2011; Wei et at., Circulation 2018, Brandao et al., PNAS 2020…)

In general, as much as possible, I would add the targets of the miRs cited across the review (in the text or in a table)

Minor points :

- Title : replace “cell” by “cells”

- line 37 : remove 3 at the beginning of the sentence after brackets

- line 80 : add “are” before express

Author Response

Rely to Reviewers 2 Comments:

Comment 1.  Page 2, lines 51 to 67: It would be interesting for the readers to add some aspects of macrophages regulation during obesity by the recently discovered family of innate lymphoid cells.

Response: We thank the reviewer for this valuable and insightful suggestion to strengthen the review article. We have added a paragraph on how innate lymphoid cells regulate obesity.

Page 2 Line 51-60

Innate lymphoid cells appeared as key element in obesity and immune response. Interestingly, ILC2 have been reported in gut mucosa and lymphoid clusters associated with fat [11]. IL-5 that is obtained by ILC2 is essential for the stimulation of eosinophil and their migration to AT, while IL-13 promote macrophage activation [12]. Together these ILC2 derived cytokines help in macrophage homeostasis and play a protective role in obesity induced metabolic disorders. Meanwhile, the deficiency of ILC2 elevates adipocity and IR in HFD- mice [13] In AT the ILC2 are elevated by IL-33 and IL-23 signals [14]. As ILC2 limit the inflammation and help to maintain metabolic homeostasis adiposeresident ILC1 promote disease progression such as IR [15]. Hence it is shown that ILC2 has protective role against obesity while ILC1 has adverse effects.

Comment 2. Page 2/3, lines 80 to 84: It could informative to mention the targets of the miRs cited by the authors and their precise role(s).

Response: We thank the reviewer for his valuable suggestion to review article. We have added the possible targets for these miRNA and cited the related article and also highlight the role of these miRNAs.

Page 3 Line 93-96.

Similarly, miR-133a,[32] miR-133b,[33] miR-208a, [34] miR-208b, miR-499,[35] miR-486,[36] and miR-1 [37] helps in proliferation and differentiation of skeletal muscles are expressed in muscle cells and are known as myomiRs [38].

Comment 3. Page 4, lines 125 to 133: I would just mention here that the role of miR and macrophages will be detailed later in the review. Is there evidence on the role of miR in other immune cell subsets during obesity-associated inflammation?

Response: We thank the reviewer for suggestion on the role of miRs and macrophages. In the revised review, we have added this during obesity-associated inflammation and cited related article.

Page 4 Line 147-149.

Beside macrophages and T-cells it is reported that B-cells miRNA also play a key role in obesity. For example, miR-150 is studied to modulates AT function by B cells activation and correlation with other immune cells [56].

Comment 4.

  1. Page 4: line 141: Please replace “hormones” with “cytokines”

Response:  In the revised review, we have corrected this oversight.

Comment 5.

Page 4 line 142: Please add example(s) with references.

Response: We thank the reviewer for this suggestion.  In the revised review, we have added the example with references.

Comment 6.

It would be interesting to summarize (and/or to describe deeper) what is known about DICER deficient mice and metabolic diseases (Mudhasani et al., J Cell Physiol 2011; Wei et al., Circulation 2018, Brandao et al., PNAS 2020).

Response: We thank the reviewer for his valuable suggestion to strengthen the review. In the revised review, we have described deeper DICER deficient mice.

Page 6 Line 216-221.

DICER is studied as main miRNA biogenesis enzyme. Recently Brandao et. al reported that exercise training stimulates the DICER in AT of both mice and human. Moreover, miR-203-3p is upregulated as result of exercise training. Together DICER-miR-203-3p is upregulated as result of exercise which ultimately help the body in metabolic function [74]. Mudhasani et al. studied the genetic role of miRNA in regulation of adipogenesis and Dicer requirement in WAT and BAT formation [75].

Comment 7.

In general, as much as possible, I would add the targets of the miRs cited across the review (in the text or a table)

Response: We thank the reviewer for his valuable and insightful suggestion to strengthen the review article. In the revised review, we have added the target of miRs in the table.

Comment 8.

Title: replace “cell” by “cells”

Response: In the revised review, we have changed as suggested by reviewers.

Comment 9.

line 37: remove 3 at the beginning of the sentence after brackets

Response: In the revised review, we have removed 3 from the sentence.

Comment 10.

.line 80: add “are” before express

Response:  In the revised review, we have corrected this mistake.  

Round 2

Reviewer 1 Report

Thank you for accepting my suggestions.